# Treatment strategies for non-responders to oral iron and folic acid treatment in anemic children: A systematic review

Rukman Manapurath[1,2], Sunita Taneja[1], Nita Bhandari[1], Ranadip Chowdhury[1]*

**1** Society for Applied Studies, Delhi, India, **2** Centre for International Health, University of Bergen, Bergen, Norway

* ranadip.chowdhury@sas.org.in

## Abstract

### Background

Iron and folic acid (IFA) are essential nutrients, with deficiencies associated not only with anemia but also with other significant health consequences, including impaired cognitive development, increased susceptibility to infections, and adverse pregnancy outcomes. Despite the widespread use of IFA for management, a notable percentage of individuals failed to respond resulting in persistent anemia. This systematic review investigates the management of non-responders to oral iron and folic acid (IFA) treatment, among children under five. Non-responders are anemic individuals who do not recover after the standard IFA treatment. A comprehensive search was conducted across multiple databases including Medline, Cochrane, Embase, and Google Scholar, covering the period from January 1, 2000, to May 31, 2024. From the initial search of 14,242 studies, we conducted title and abstract screening, and 27 articles were selected for full text screening. After further exclusion, a total of 8 studies were identified, including randomized controlled trials, cohort studies, and case series. The review found that intravenous management, particularly ferric carboxymaltose, was found to be effective in cases of iron non-responsiveness. However, the causes of poor/non-responders to oral iron are less explored, indicating a need for further research. The review also identified a lack of high-quality studies on this topic. The review highlights the limited evidence on managing anemia unresponsive to oral iron, especially in low- and middle-income countries. While intravenous iron shows promise, more data is required to draw solid conclusions. Developing personalized treatment strategies is crucial to improving outcomes and addressing the global burden of anemia.

## Introduction

Anemia, a global health issue affecting both developing and developed nations, has multifactorial etiologies. Among these, nutritional deficiencies—particularly iron and folic acid (IFA) deficiencies—are critical contributors to anemia, especially in vulnerable populations such as children [1]. Iron is essential for the production of hemoglobin, a protein in red blood cells

**Data availability statement:** All data used in this systematic review is available as published, manuscript in public domain.

**Funding:** This study was supported by the Indian Council of Medical Research (ICMR), Extramural Fund under grant number 5/7/594/1/-RHN to RC (Ranadip Chowdhury). The funders had no role in the study design, data collection and analysis, decision to publish, or preparation of the manuscript.

**Competing interests:** The authors have declared that no competing interests exist.

that transports oxygen from the lungs to the body's tissues. Folic acid is crucial for DNA synthesis and cell division, especially during periods of rapid growth such as fetal development and early childhood [2]. Deficiencies can lead to anemia, characterized by a reduction in the number of red blood cells or hemoglobin in the blood. This condition may result in impaired cognitive development and an increased risk of maternal and child mortality. According to the World Health Organization, 40% of all children aged 6–59 months are affected by anemia, a substantial portion of which is attributable to nutritional causes [3].

Oral IFA supplementation has been extensively adopted to populations at risk, such as young children [4]. While this has proven to be effective in reducing deficiencies in many cases, there remains a proportion of individuals who do not respond to management with IFA (ranging from 30 to 60%, depending on the duration of treatment) [5,6]. These non-responders are defined as anemic children who do not achieve a clinically significant increase in hemoglobin levels after the standard IFA regimen. This lack of response could be due to several factors, such as inflammation, infections, poor compliance, and genetic disorders (e.g., thalassemia and sickle cell disease) affecting iron metabolism [7].

In instances where anemia remains unresponsive to IFA management, it becomes necessary to explore alternative treatment approaches. These alternative management interventions range from intravenous iron therapy and blood transfusions to dietary modifications and supplementation with other potentially deficient micronutrients that may be lacking. This systematic review aims to review such treatment strategies among non-responders to IFA treatment in children, particularly under five years of age. This question is of critical importance in the field of global health. Effective management of non-responders to IFA treatment is key to reducing anemia burden, allowing for targeted interventions and preventing disease escalation to severe stages. This approach decreases the health and economic burden of severe anemia and contributes to a more accurate and potentially lower anemia prevalence rate.

## Methods

In accordance with the PRISMA Checklist, this systematic review was conducted to identify pertinent studies on anemia treatment strategies for non-responders to IFA supplementation among children. Non-responders were defined as anemic individuals who did not recover after the standard IFA regimen or as defined by the study authors.

### Search strategy

The search strategy was designed to access a broad range of studies without language restrictions. The electronic databases used for the search included Medline, Cochrane, Google Scholar, and Embase. These databases were chosen due to their extensive coverage of medical and health-related research literature, covering the period from January 1, 2000, to May 31, 2024 (See S1 Table).

The search terms were carefully selected to capture the key concepts of the research question. The MeSH term "Anemia/epidemiology" [MAJR] was used to identify studies on the epidemiology of anemia. This was combined with the term "Government Programs" [MAJR] AND "micronutrient" [tw] to find studies related to micronutrient supplementation programs for anemia. To specifically target the population of interest, i.e., non-responders to IFA treatment, the following keywords were used: "treatment-refractory" [tw], refractory [tw], treatment-resistan* [tw], failure [mh], and "non-responder" [tw]. The use of these terms ensured that the search results included studies on individuals who did not respond to standard IFA treatment.

### Study selection and eligibility criteria

This review included studies focusing on anemic children who did not respond to oral IFA supplementation. Studies were considered eligible if they met the criteria of involving these specific populations, examining treatment strategies for non-responders to IFA treatment, and reporting outcomes such as changes in hemoglobin levels, serum ferritin, or other relevant hematological parameters. Eligible study types included observational studies (such as cohort, case-control, and cross-sectional studies), randomized controlled trials (RCTs), and non-randomized controlled trials. The interventions of interest were alternative treatment strategies for non-responders to IFA.

Studies that involved non-responders in populations where anemia is secondary to other conditions, such as chronic renal failure, inflammatory bowel disease, or genetic disorders like thalassemia and sickle cell anemia, were excluded. These conditions often require specialized management beyond nutritional interventions, which falls outside the scope of this review's focus on nutritional anemia.

### Data extraction

Search results from each database were imported into Covidence reference management software [8], and duplicates were removed. Screening and full-text review of the articles were managed using the same software. Data from the included studies were extracted using a standardized data extraction form by two independent reviewers. The extracted data included study characteristics (e.g., author, year, country), study design, population characteristics, details of the intervention and comparator, outcome measures, and key findings. The quality of the included studies was assessed using appropriate risk of bias tools. The data were then synthesized narratively. A meta-analysis could not be conducted due to the diverse nature of the included studies. Additionally, the varied strategies employed by different studies to handle poor or non-responders further complicated the feasibility of a meta-analysis.

### Assessment of quality

The risk of bias in the included studies was assessed using the Newcastle–Ottawa Scale (NOS) for observational studies. The NOS scores were categorized as high risk of bias (total score of 0–2), moderate risk of bias (total score of 3–5), and low risk of bias (total score of 6–8) [9]. For RCTs, the Cochrane Risk of Bias (ROB) tool was utilized to provide a more suitable quality assessment [10]. Case series were assessed using the Joanna Briggs Institute (JBI) tool for case series to determine their quality [11]. Two review authors (R M. and R.C.) independently assessed the risk of bias.

## Results

The systematic review yielded 14,242 studies from the initial search across Medline, Cochrane, Google Scholar and Embase. After removing duplicates, 13,495 studies remained. The title and abstract screening excluded 13,468 studies, primarily because they did not focus on anemia treatment in non-responders to oral iron. The full-text assessment of the remaining 27 studies resulted in a further exclusion of 20 articles (See S2 Table), and 7 studies met the inclusion criteria and were included in the review (Fig 1). The included studies were conducted in various countries, with a significant proportion from low- and middle-income countries where anemia is a major public health issue. The studies varied in design, including RCTs, cohort, and cross-sectional studies. Refer to Table 1 for characteristics of the included studies.

The review included studies conducted in various countries, including India, Greece, USA, Switzerland, Bangladesh, and Italy. The studies varied in type, with cohort studies,

**Fig 1. Flow diagram of study selection process.**

randomized controlled trials (RCTs), and case series. The sample sizes ranged from 87 to 260 participants in cohorts and RCT. The ages of the participants varied widely, with some studies focusing on children under 5 years of age, while others included participants up to 18 years of age. The studies primarily included participants who failed to respond to oral iron and excluded those with chronic diseases such as blood loss conditions or kidney diseases. The outcomes generally involved an increase in hemoglobin levels, with some studies also reporting increases in serum iron or ferritin levels. The included studies varied in their definitions of non-response to IFA treatment. Generally, non-response was described as the failure to achieve a clinically meaningful increase in hemoglobin levels after a standard treatment period, which typically ranged from 8 to 12 weeks. Specific definitions and follow-up durations for each study are detailed in S3 Table.

The studies incorporated a broad spectrum of interventions to address iron deficiency. The route of administration varied between oral and intravenous (IV) formulations. IV iron was administered in different regimens, including iron sucrose, ferric carboxymaltose (FCM), and low molecular weight iron dextran (LMWID). IV iron was typically given to correct the

**Table 1. Characteristics of included studies.**

| Study Type | Study design | Sample size, Type of anemia, Age group | Definition of failure to respond | Follow-up Duration Range | Intervention | Outcomes |
|---|---|---|---|---|---|---|
| Powers 2015, 2017 [12,13] | Retrospective cohort | 72, mild anemia, median age 14 yrs (range 9 months to 20.8 years) | Taken oral iron therapy for a median of 4 months (IQR, 2-12 months) yet exhibited limited or no increase above their baseline Hgb concentration | 12 weeks | INTRA VENOUS - Ferric carboxymaltose (~435 mg) single dose | Pre Hb-9.3 – post Hb 12.3 g/dL |
| Ozsahin 2020 [14] | Retrospective cohort | 144, 18 mon to 18 yrs, mild anemia | Failure of oral iron therapy (not explicitly defined) | 6 to 12 weeks | INTRA VENOUS - Ferric carboxymaltose 20mg/kg in a single session | <6 yrs: 85% serum ferritin>30 micro g/L |
| Plummer 2013 [15] | Case series | 31, 11 to 18 yrs, mild to moderate anemia | Failed to respond to or not felt to be candidates for oral iron. | 4 to 16 weeks | INTRA VENOUS - LMW Iron dextrose over 60 min in OPD | 11% increase in mean Hb compared to baseline |
| Crary 2011 [16] | Case series | 13, 3 mon to 18 yrs, mild to moderate anemia | | Not defined | INTRA VENOUS - Iron sucrose ranging from 25 to 500 mg over median dosing of 3 times | For children with refractory IDA, Median Hb rise of 3.1 g/dL (0.8, 7.6) compared to baseline |
| Sarker 2008 [5] | RCT | 260, under 5, mild anemia | | 90 days | Gp1- Anti H pylori Rx with iron, Gp2 –Anti Hp alone Gp 3- Oral iron 90 days Gp 4- Placebo | Changes in Hb: Gp1- 9.6 to 11.2 g/dL Gp2 –9.7 to 10.4 g/dL Gp 3- 9.6 to 11.3 g/dL Gp 4- 9.5 to 10.3 g/dL (Hpylori not likely to cause treatment failure of oral Fe) |
| Giovanna Russo 2016 [17] | Observational, prospective, multicenter | 107, IDA, 3 months to 12 years | | 24 weeks | Various oral iron formulations: ferrous gluconate/ sulfate (2 or 4 mg/kg), ferric iron salts, bis-glycinate iron, liposomal iron | Median increase in hemoglobin levels at 2 and 8 weeks, reticulocyte increase at 3 days, gastrointestinal side effects (16% for ferrous salts, 6% for bis-glycinate) |

IDA- Iron Deficiency Anemia; Gp- Group.

total iron deficit, with dosing schedules ranging from every other day (in the case of iron sucrose) to single or two-session doses (in the case of FCM and ferumoxytol). The duration of follow-up ranged from 6 to 12 weeks for most IV interventions. In contrast, oral iron formulations, such as ferrous salts, bis-glycinate iron, and liposomal iron, were administered daily over longer durations, with one study involving a follow-up period of 24 weeks. One study specifically targeted H. pylori-infected children and tested different anti-H. pylori therapies combined with oral iron, with a 90-day follow-up period. Overall, the diversity in the interventions and follow-up durations provided a comprehensive understanding of treatment responses in both oral and IV iron formulations across different populations. See S3 Table for detailed data extraction table.

## Response to oral iron formulations

Two studies reported the effectiveness of various oral iron formulations in managing non-responsiveness to IFA. One RCT addressed the underlying infection (H. pylori). However, this did not improve iron absorption [5]. In another study, switching to more bioavailable and better-tolerated forms of iron (like bis-glycinate and liposomal iron) likely contributed to improved outcomes in children who had initially failed to respond to traditional ferrous salt formulations [17].

### Response to intra venous (IV) formulations

Research conducted with Low Molecular Weight Dextran demonstrated an enhancement in hemoglobin levels in children who showed resistance to oral iron treatment. However, mild side effects were observed in 30% of the cases [15]. IV sucrose was used in three studies refractory to oral iron treatment and found to improve outcomes in cases of malabsorption [16,18,19].

The utilization of FCM in anemia treatment has demonstrated promising outcomes. A retrospective cohort study involving 72 patients with mild anemia (median age 14 years), administering a single dose of IV FCM (~435 mg) reported a significant increase in hemoglobin levels from 9.3 to 12.3 g/dL over a 12-week follow-up period [12,13]. In a similar retrospective cohort study, 144 patients (aged 18 months to 18 years with mild anemia) were treated with a single session of IV FCM (20 mg/kg) [14]. The follow-up duration ranged from 6 to 12 weeks, and for patients under 6 years, 85% had serum ferritin levels greater than 30 micro g/L. In a RCT, 477 women of reproductive age (only 60% were refractory to oral iron) with mild anemia were treated with IV FCM (1000 mg over 15 min) or oral iron (65 mg elemental iron thrice daily for 6 weeks). The results showed that 82% of the FCM group achieved a hemoglobin rise by 2 g/dL or more within 42 days of treatment, compared to 61.8% of the oral iron group [6]. These findings suggest that FCM can be an effective intervention for treating different types of anemia across various age groups.

In the study by Sarkar et al., iron supplementation was monitored for 90 days, with adherence ensured through supervised iron administration. Similarly, Russo et al. (2020) recorded adherence by engaging parents during follow-up visits to assess compliance and report any side effects. However, other studies included in the review did not systematically assess compliance.The included studies used varying methods to assess anemia, with some explicitly using venous blood samples (e.g., Sarker 2008, Russo 2020) while others did not specify the sampling technique (e.g., Crary 2011). In studies that included measurements of serum ferritin or transferrin saturation (e.g., Powers 2015, Powers 2017, Ozsahin 2020), venous blood sampling was likely used. Definitions of anemia were generally based on hemoglobin concentration thresholds, often aligning with WHO criteria, and some studies incorporated additional parameters such as ferritin or transferrin saturation levels to confirm iron deficiency.

Most of the studies included were of moderate-to-low risk of bias (see S4 Table for bias assessment).

As this review is narrative in nature, no issues related to missing data were encountered. All relevant data reported in the included studies were available for analysis and synthesis. As such, no imputation methods or sensitivity analyses were required.

## Discussion

This systematic review, the first of its kind focusing on the management of non-responders to oral iron therapy in vulnerable groups, has provided valuable insights into the treatment of anemia. The review encompassed a diverse range of studies conducted in various countries, with a significant proportion from low- and middle-income countries where anemia is a major public health issue. The studies varied in design, including RCTs, cohort studies, and cross-sectional studies, and the interventions used demonstrated a variety of treatment approaches for iron deficiency. The findings suggest that tailored treatments based on individual needs and conditions can be effective in managing non-responsiveness to IFA. The outcomes of these interventions are primarily measured in terms of hemoglobin (Hb) levels, with some studies also considering serum ferritin levels. The majority of the studies reported an increase in Hb levels post-intervention, indicating the effectiveness of the treatments.

However, the degree of improvement varied across studies, possibly due to differences in the severity of anemia, the specific type of anemia, the intervention used, and the age group of the participants.

In terms of treatment strategies, a common theme across the studies was the need for alternative treatment regimens, such as the use of parenteral iron or combination therapy with other micronutrients like vitamin C. The type of iron supplement varied across studies, with some using ferrous ascorbate, iron sucrose, ferric carboxymaltose, or ferrous bis-glycinate chelate. Some studies also incorporated vitamin C or anti-H.pylori treatment in their intervention strategies. The effectiveness varied depending on the severity of anemia and patient characteristics. These findings underscore the importance of exploring tailored approaches to address non-responsiveness to oral iron.

This review shows that IV management is superior in cases of iron non-responsiveness. Based on available evidence, we can infer that FCM brings a rapid response to anemia or rise in hemoglobin for those non-responsive to oral iron. It is also superior to IV ferrous sucrose as the former requires single-dose administration for observable benefits and shorter administration time. Overall, IV sucrose and IV FCM were both effective in addressing anemia refractory to oral iron. IV sucrose was administered to younger children and required multiple dosing sessions, primarily addressing malabsorption-related anemia. In contrast, IV FCM was predominantly used in older children and adolescents. More studies that exclusively enrol non-responsive cases and more side effects, if any, need to be studied, especially in pregnant women. Also, every case should be followed up individually to identify the causes of anemia. An algorithm needs to be developed for managing non-responsiveness to oral iron. The most critical gap is the non-availability of a comprehensive understanding of the causes of poor/non-responders of oral iron. This needs to be studied further.

Non-responders who do not show a significant increase in hemoglobin levels post-treatment pose a critical challenge in anemia management. It is crucial to understand the reasons for this lack of response, which could range from individual physiological differences to the presence of other underlying conditions. However, current studies and clinical trials have not adequately addressed this issue. One of the major reasons for non-responsiveness to oral iron therapy in the context of nutritional anemia is the poor adherence to treatment. Non-compliance is a widespread issue, particularly in low-resource settings, where factors such as gastrointestinal side effects, the unpleasant taste of iron supplements, and low patient follow-up lead to reduced effectiveness.

This lack of focus on non-responders is a significant limitation, as it hinders the development of comprehensive treatment protocols that can cater to all patient categories. Therefore, there is a growing need for precision medicine approaches to manage individual conditions. The exploration of new compounds and treatment regimens could lead to the discovery of more effective treatments for non-responders. A significant strength of this review is that it is the first of its kind to focus on the management of non-responders to oral iron therapy in vulnerable groups, offering valuable insights through a diverse range of studies from various countries and advocating for the effectiveness of tailored treatments based on individual needs and conditions. Further research is needed focusing on non-responders to iron therapy in low- and middle-income countries to ensure that findings are relevant and applicable to the regions most affected by anemia.

## Limitations

Several limitations of this review should be noted. First, the heterogeneity of the included studies, in terms of both study design and intervention types, limits the ability to draw firm conclusions regarding the effectiveness of specific treatments. Moreover, there was a lack

of consistency in defining non-responsiveness across studies, with some studies measuring different outcomes (e.g., hemoglobin levels, transferrin saturation). Finally, the review did not include enough data from long-term follow-up studies, making it difficult to assess the sustained effectiveness of the interventions.

In conclusion, this systematic review has shed light on the critical issue of non-responders to oral iron and folic acid treatment, particularly among women and children under five. The diverse range of interventions used in the studies underscores the need for personalized treatment strategies based on individual needs and conditions. The review highlights the superiority of intravenous management, particularly FCM, in cases of iron non-responsiveness. However, the lack of a comprehensive understanding of the causes of poor/non-responders to oral iron necessitates further research.

## Supporting information

**S1 Table. Search strategy: Detailed search strategy used for identifying relevant studies from databases like medline, EMBASE, and Google Scholar.**
(DOCX)

**S2 Table. List of excluded articles and reasons for exclusion.**
(DOCX)

**S3 Table. Detailed data extraction table: Extracted data from all studies included in the systematic review.**
(DOCX)

**S4 Table. (a) Risk of bias assessment (Newcastle–Ottawa Scale).** (b) Summary of Cochrane Risk of Bias (RoB) for Randomized Controlled Trials. (c) JBI Critical Appraisal for Case Series.
(DOCX)

**S5 Table. List of studies included in the review.**
(DOCX)

AcknowledgmentOur gratitude is extended to Dr. Anju Sinha, MD, PhD, a Scientist Consultant at the Division of Reproductive, Maternal, Newborn, Child and Adolescent Health, Indian Council of Medical Research Headquarters, for her invaluable support throughout this process.

**Other information:** This systematic review was commissioned by the Indian Council of Medical Research, New Delhi, India for the purpose of generating evidence on the management protocol of anemia in vulnerable groups. The protocol for this review has been officially registered with PROSPERO, the international prospective register of systematic reviews. The registration number is CRD42024552200.

## Author contributions

**Conceptualization:** Rukman Manapurath, Sunita Taneja, Nita Bhandari, Ranadip Chowdhury.

**Data curation:** Rukman Manapurath, Sunita Taneja, Ranadip Chowdhury.

**Formal analysis:** Rukman Manapurath, Ranadip Chowdhury.

**Funding acquisition:** Rukman Manapurath, Sunita Taneja, Nita Bhandari, Ranadip Chowdhury.

**Investigation:** Rukman Manapurath, Ranadip Chowdhury.

**Methodology:** Rukman Manapurath, Sunita Taneja, Ranadip Chowdhury.

**Project administration:** Rukman Manapurath, Sunita Taneja, Nita Bhandari, Ranadip Chowdhury.

**Software:** Rukman Manapurath, Ranadip Chowdhury.

**Supervision:** Sunita Taneja, Nita Bhandari, Ranadip Chowdhury.

**Validation:** Sunita Taneja, Nita Bhandari.

**Visualization:** Ranadip Chowdhury.

**Writing – original draft:** Rukman Manapurath, Ranadip Chowdhury.

**Writing – review & editing:** Rukman Manapurath, Sunita Taneja, Nita Bhandari, Ranadip Chowdhury.

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
