## [Decision Letter · Decision Letter 0]

27 Aug 2024

PGPH-D-24-01359

Effective Treatment Strategies for Women and Children Under Five: Addressing Non-Responders to Oral Iron and Folic Acid- A Systematic Review

Dear Dr. Chowdhury,

Thank you for submitting your manuscript to PLOS Global Public Health. After careful consideration, we feel that it has merit but does not fully meet PLOS Global Public Health’s publication criteria as it currently stands. Therefore, we invite you to submit a revised version of the manuscript that addresses the points raised during the review process.

We look forward to receiving your revised manuscript.

Kind regards,

Academic Editor

Journal Requirements:

Additional Editor Comments (if provided):

The authors have dealt with an important topic of non-response to oral iron. However, there are methodological weaknesses in the review and the scope of the topic is not clear. Considering the burden of nutritional anemia, the topic should be restricted to nutritional anemia and exclude other studies related to genetic disorders.

Following are few comments and suggestions:

Title: The title should mention clearly that the manuscript describes evidence on non-responders to anemia. Presently the title is broken into three parts and reads a bit awkward.

Abstract: The search strategy mentioned in the abstract is different from what is mentioned in the methods section. Abstract mentions Cochrane database which is not mentioned in the methods. Also the start date for search is not clear in the methods.

The abstract should be in structured format.

Introduction:

- The description should specify the focus on nutritional anemia as there are other etiologies of anemia which may or may not benefit from IFA supplementation.

- What is the definition of non-responder to IFA in WRA and children? Not clear.

- Line 57-58 – one of major reason for non-response to IFA could be other etiologies of anemia such as hemoglobinopathies. In this case, establishing the etiology for non-response would be more important. The authors have not elaborated on this aspect.

- The compliance to oral iron is often suboptimal leading to reduced effectiveness. Is this aspect considered in the identification of non-responders.

Methods:

- Details regarding the type of reference software used not mentioned.

- The methods section does not describe the population covered, type of studies, inclusion exclusion criteria for studies to be included. Also it is not clear whether studies related to non-responders in other disease populations with anemia such as chronic renal failure, inflammatory bowel disease, thalassemia were to be excluded from the manuscript.

- New Castle Ottawa scale – This scale can be used for cohort studies. However, for RCTs the authors should have used ROB or other relevant tool. The authors have not assessed the quality of case series although it can be assessed through tools such as JBI tool (https://jbi.global/sites/default/files/2020-07/Checklist_for_Case_Series_0.pdf).

- It is not clear why the authors have not considered reviews or pooled analysis data in this review as there is limited evidence in terms of original studies. E.g. Okam MM, Koch TA, Tran MH. Iron deficiency anemia treatment response to oral iron therapy: a pooled analysis of five randomized controlled trials. Haematologica. 2016 Jan;101(1):e6-7. doi: 10.3324/haematol.2015.129114. This is a very relevant paper which has not been included.

Results

Table 1: It combines many variables into one column. Suggest to keep them separate for better clarity. Definition of non-response in each of these studies should be included in the table.

- Details of how many studies in women and children should be included in the description.

- The search results are questionable – The authors have included about 7-8 studies in pediatric population. However, only couple of studies have been included in women of reproductive age and no studies in pregnant women have been included. Case series on IRIDA are included in the results. However, IRIDA is a rare genetic disorder of iron which does not contribute to major burden of anemia. Ideally, these should have been excluded.

- It is not clear, why the authors have included a study on celiac disease if other causes of anemia like chronic renal disease are excluded. Also, was this a study in WRA or children?

- The authors have combined results from all age group which should be presented separately.

Discussion and conclusion

- The authors have not elaborated on the major reasons for non-responders to oral iron in the setting of nutritional anemia which is at the heart of this topic.

- Non-compliance to oral iron could be a factor responsible for non-response. The discussion does not cover this. Non-compliance to oral iron is a major public health issue.

- The authors need to elaborate on why in some studies the participants responded to oral iron after initial failure – was it because of high dose or presence of ascorbic acid?

- Finally the authors have discussed about the need for precision medicines which is a generic truth. The purpose of this review is to collate the available evidence to provide consensus.

- The authors have not discussed about the best available evidence in specific populations like WRA and children.

- The authors do not describe limitations of their review.

Reviewers' comments:

Reviewer's Responses to Questions

**Comments to the Author**

1. Does this manuscript meet PLOS Global Public Health’s publication criteria ? Is the manuscript technically sound, and do the data support the conclusions? The manuscript must describe methodologically and ethically rigorous research with conclusions that are appropriately drawn based on the data presented.

Reviewer #1: Yes

Reviewer #2: No

2. Has the statistical analysis been performed appropriately and rigorously?

Reviewer #1: Yes

Reviewer #2: N/A

3. Have the authors made all data underlying the findings in their manuscript fully available (please refer to the Data Availability Statement at the start of the manuscript PDF file)?

Reviewer #1: Yes

Reviewer #2: No

4. Is the manuscript presented in an intelligible fashion and written in standard English?

Reviewer #1: Yes

Reviewer #2: No

5. Review Comments to the Author

Reviewer #1: This is a very important study that systematically identifies non-responders to IFA among high-risk populations in society, and effective treatment regimen for the non-responders. It also serves as the basis for future studies.

Abstract:

It will be more appropriate if the abstract is grouped or differentiated into background, methods, results, and conclusion

Introduction:

The Introduction gives the background information about non-responders to IFA treatment. Is IFA used for the treatment of all categories of anaemia (mild, moderate, severe)? Hence, the authors should tell the IFA regimen for the type of anaemia in this section.

Methods:

Study selection and eligibility criteria

Line 87, could the authors mention the name of the reference management software program for reproducibility purposes?

Assessment of quality

From line 113, why did the authors combine published articles of qualitative nature (case series, case reports) and that of quantitative strand (RCT, cross-sectional) in this systematic review?

Discussion:

The discussion of the treatment regimen for non-responders to IFA in developing countries should be enhanced since the authors mentioned that anaemia is very prevalent in developing countries.

Reviewer #2: The reviewer appreciates the authors for working on an important public health topic ‘anemia’. However, the manuscript suffers from major methodological inadequacies and weaknesses, that render it unsuitable for publication.

Title: The first half of the title is vague. The term anemia is not reflected in the title.

Methods: Why was the search restricted to two search engines? RCTs have been included in the review but clinical trials.gov was not searched.

A systematic review aims to collect selective published literature on a specific condition to arrive at the direction for the management of a condition. Although anemia is chosen, a well defined selection criteria for the type of anemia, its severity, and the study setting, type of iron, minimum saple size (clinical or population) would have improved the quality of the selected studies and minimized heterogeneity . Thus poor selection criteria of the studies have contributed to vague findings.

For instance, how can a ‘unique scenario” as described by the authors, a genetic disorder form one case study be compiled together with population-based studies with over 400 samples or RCTs?

A well-defined selection criteria would have solved many of the methodological weaknesses.

The quality assessment tool assesses the quality of non-randomized studies. However, the selection included RCTs.

Table 1: Hb improved from 20-70% is vague. Anemia prevalence decreased or hemoglobin status improved among certain percent etc.

Table 1: How can treatment for mild anemia be compared with a genetic condition or infectious condition?

Anemia is a public health problem in developing countries, but studies from Italy, USA and Switzerland have been used for comparison.

The findings of the SR do not contribute to any novel findings: Iron with ascorbate, IV management, or blood transfusion are previously proven strategies.

6. PLOS authors have the option to publish the peer review history of their article (what does this mean? ). If published, this will include your full peer review and any attached files.

**Do you want your identity to be public for this peer review?** For information about this choice, including consent withdrawal, please see our Privacy Policy .

Reviewer #1: No

Reviewer #2: No

---

## [Decision Letter · Decision Letter 1]

29 Dec 2024

PGPH-D-24-01359R1

Effective Treatment Strategies for Non-Responders to Oral Iron and Folic Acid Treatment in Anemic Children: - A Systematic Review

Dear Dr. Chowdhury,

Thank you for submitting your manuscript to PLOS Global Public Health. After careful consideration, we feel that it has merit but does not fully meet PLOS Global Public Health’s publication criteria as it currently stands. Therefore, we invite you to submit a revised version of the manuscript that addresses the points raised during the review process.

We look forward to receiving your revised manuscript.

Kind regards,

Paraskevi Detopoulou

Academic Editor

Reviewers' comments:

Reviewer's Responses to Questions

**Comments to the Author**

1. If the authors have adequately addressed your comments raised in a previous round of review and you feel that this manuscript is now acceptable for publication, you may indicate that here to bypass the “Comments to the Author” section, enter your conflict of interest statement in the “Confidential to Editor” section, and submit your "Accept" recommendation.

Reviewer #3: (No Response)

Reviewer #4: (No Response)

2. Does this manuscript meet PLOS Global Public Health’s publication criteria ? Is the manuscript technically sound, and do the data support the conclusions? The manuscript must describe methodologically and ethically rigorous research with conclusions that are appropriately drawn based on the data presented.

Reviewer #3: Yes

Reviewer #4: Yes

3. Has the statistical analysis been performed appropriately and rigorously?

Reviewer #3: I don't know

Reviewer #4: I don't know

4. Have the authors made all data underlying the findings in their manuscript fully available (please refer to the Data Availability Statement at the start of the manuscript PDF file)?

Reviewer #3: Yes

Reviewer #4: Yes

5. Is the manuscript presented in an intelligible fashion and written in standard English?

Reviewer #3: No

Reviewer #4: Yes

6. Review Comments to the Author

Reviewer #3: This manuscript describes a systematic review that aims to investigate non-responsiveness to iron and folic acid supplementation in anemic children.

Queries and comments to be addressed:

Why only include anemic children? It would be relevant to assess non-responsiveness to IFA in both anemic and non-anemic populations for increased generalizability (and iron deficient and replete, as well).

Further, i think it is worth comment on how anemia was assessed across these eligible studies. Single-drop capillary blood has shown to render highly inaccurate estimations of hemoglobin concentration. Was single-drop or venous blood used, and how was anemia defined?

It is not clear why the title includes 'effective treatment strategies' as i do not think this can be confirmed from the scope and content of this review.

Was the review registered in any online repository, such as PROSPERO? Were there any deviations from the initial protocol?

Line 86-93: the scope of the search seems limiting, e.g., use of several very specific key word searches. Did the authors seek help from a librarian or expert in literature searches to execute the search?

Abstract: Suggest to revise first sentence as it implies the only consequence of IFA deficiency is anemia; when in fact, there are numerous and some of which are more important than anemia. Suggest to replace the word 'suffer' with more objective text. Please clarify what is meant by 'standard IFA treatment'. Authors mention case series (do they mean case studies) or is clarification required here? No need to capitalize the word 'intravenous'.

Please add more detail to tool(s) used assess the quality of bias?

Line 107: with what software?

Line 78: was this assessed within a specific timeframe? when was 'non-response' assessed?

Adherence to the IFA is obviously a major potential confounder to the non-response outcome. Please expand upon how this was assessed across the different studies and if authors feel this was comprehensive enough to tease out the effect from these two variables.

Reviewer #4: Many thanks to the Editor for the opportunity to review this paper. The topic is relevant, and anemia is a growing concern for the vulnerable groups, especially in LMICs. Overall, the paper is well written but i have minor concerns that will be helpful to the authors.

The search terms did not include 'Children', or nutritional anaemia, which are part of the topic. Like, refractory nutritional anaemia in children, to align with research topic and study population. This will help provide more papers to analyze from.

MINOR:

1. There needs to be a review for typos or grammar as seen in line 64, 84, and elsewhere in the paper.

2. In line 59, Oral IFA need to be explicitly qualified. Is it supplementation?

3. In line 122, JBI should be defined.

RESULTS:

IV sucrose Vs IV FCM, it will be useful to highlight how the two patient groups differ from the various studies-

4. The too many exclusions, might be due to search terms not been specific and align to topic

DISCUSSION:

5. line 212 does not make a sense or is incomplete.

7. PLOS authors have the option to publish the peer review history of their article (what does this mean? ). If published, this will include your full peer review and any attached files.

**Do you want your identity to be public for this peer review?** For information about this choice, including consent withdrawal, please see our Privacy Policy .

Reviewer #3: No

Reviewer #4: No

---

## [Editor Report · Decision Letter 2]

3 Feb 2025

Treatment Strategies for Non-Responders to Oral Iron and Folic Acid Treatment in Anemic Children: - A Systematic Review

PGPH-D-24-01359R2

Dear Dr. Chowdhury,

We are pleased to inform you that your manuscript 'Treatment Strategies for Non-Responders to Oral Iron and Folic Acid Treatment in Anemic Children: - A Systematic Review' has been provisionally accepted for publication in PLOS Global Public Health.

Best regards,

Paraskevi Detopoulou

Academic Editor

The authors have substantially upgraded the manuscript according to reviewers' comments.